# Epistatic mutations in PUMA BH3 drive an alternate binding mode to potently and selectively inhibit anti-apoptotic Bfl-1

Justin M Jenson[1], Jeremy A Ryan[2], Robert A Grant[1], Anthony Letai[2], Amy E Keating[1,3]*

[1]Department of Biology, Massachusetts Institute of Technology, Cambridge, United States; [2]Department of Medical Oncology, Dana-Farber Cancer Institute, Boston, United States; [3]Department of Biology, Department of Biological Engineering, Massachusetts Institute of Technology, Cambridge, United States

**Abstract** Overexpression of anti-apoptotic Bcl-2 family proteins contributes to cancer progression and confers resistance to chemotherapy. Small molecules that target Bcl-2 are used in the clinic to treat leukemia, but tight and selective inhibitors are not available for Bcl-2 paralog Bfl-1. Guided by computational analysis, we designed variants of the native BH3 motif PUMA that are > 150-fold selective for Bfl-1 binding. The designed peptides potently trigger disruption of the mitochondrial outer membrane in cells dependent on Bfl-1, but not in cells dependent on other anti-apoptotic homologs. High-resolution crystal structures show that designed peptide FS2 binds Bfl-1 in a shifted geometry, relative to PUMA and other binding partners, due to a set of epistatic mutations. FS2 modified with an electrophile reacts with a cysteine near the peptide-binding groove to augment specificity. Designed Bfl-1 binders provide reagents for cellular profiling and leads for developing enhanced and cell-permeable peptide or small-molecule inhibitors.

*For correspondence: keating@mit.edu

**Competing interests:** The authors declare that no competing interests exist.

## Introduction

Anti-apoptotic members of the Bcl-2 family are broadly recognized as promising cancer therapeutic targets. Human anti-apoptotic proteins Bcl-2, Bcl-$x_L$, Bcl-w, Mcl-1 and Bfl-1 have a globular, helical fold and function by binding to short, $\alpha$-helical Bcl-2 homology 3 (BH3) motifs in pro-apoptotic proteins, as shown in *Figure 1A*. Competition for binding among BH3-containing proteins regulates mitochondrial outer membrane permeabilization (MOMP), which is an irreversible step toward caspase activation and cell death. The appropriate balance of interactions between pro-survival and pro-death Bcl-2 family members in healthy cells is often disrupted in cancer cells, where overexpression of anti-apoptotic Bcl-2 proteins can promote oncogenesis and confer resistance to chemotherapeutic agents (*Opferman, 2016*).

There has been considerable progress developing BH3 mimetic peptides and small molecules to inhibit the function of anti-apoptotic Bcl-2 proteins by blocking their interactions. One outstanding example is the small molecule venetoclax, which targets Bcl-2 and was recently approved by the FDA for treatment of chronic lymphocytic leukemia (*Souers et al., 2013*; *Roberts et al., 2016*). A major challenge in developing venetoclax was achieving specificity, which is important because Bcl-2 family members support survival of healthy cells. For example, the small molecule ABT-263 inhibits both Bcl-2 and Bcl-$x_L$, but Bcl-$x_L$ cross-reactivity leads to dose-limiting thrombocytopenia (*Rudin et al., 2012*; *Roberts et al., 2012*; *Schoenwaelder et al., 2011*). In the laboratory, highly selective inhibitors of anti-apoptotic proteins are used for profiling experiments that can establish which anti-apoptotic proteins are essential for cancer cell survival in individual patients and predict

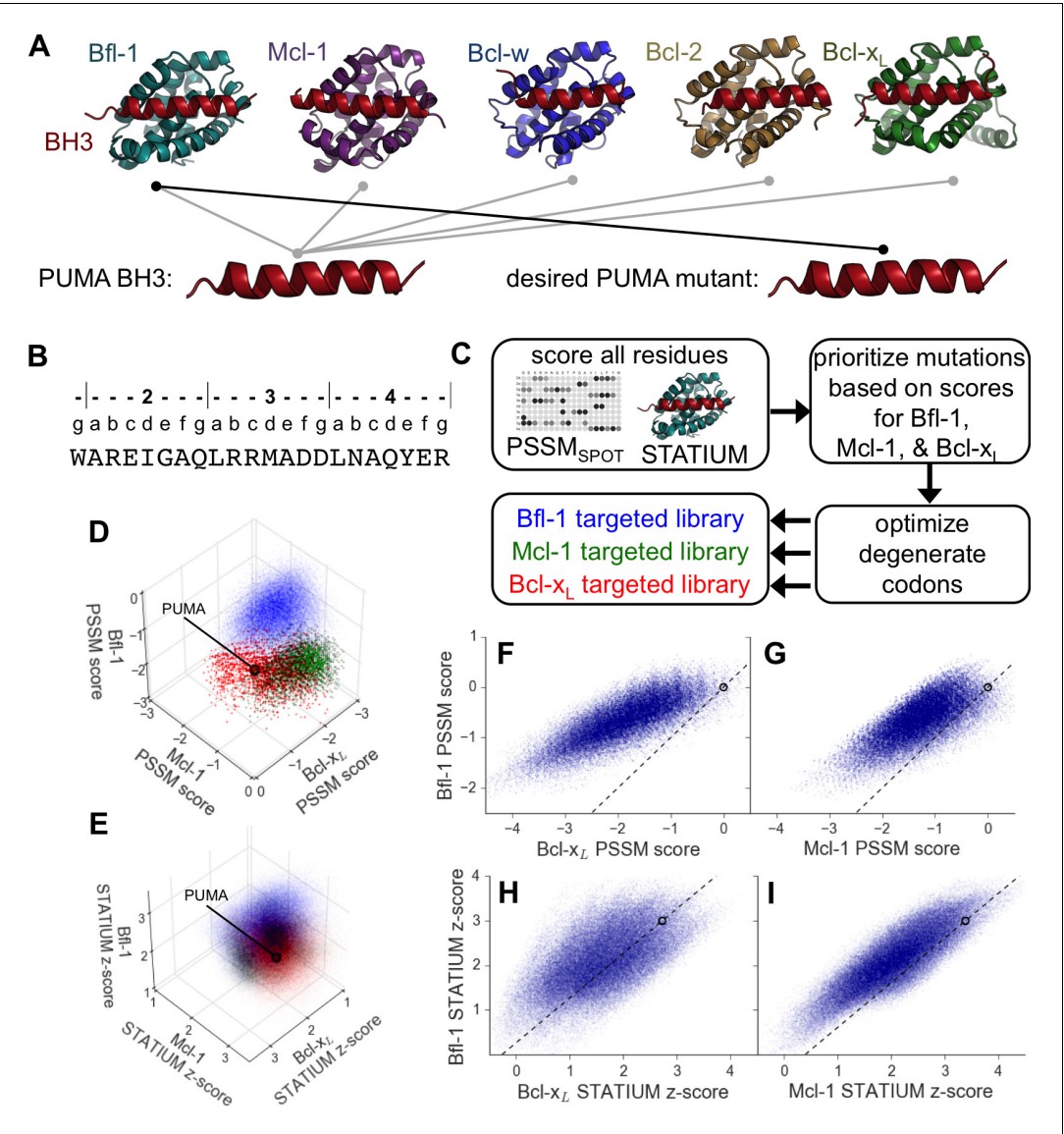

**Figure 1.** Computational design of a library of PUMA BH3 variants selective for Bfl-1. (A) PUMA BH3 is pan-selective; the design objective was a peptide that binds tightly only to Bfl-1. (B) Sequence of PUMA BH3 showing the heptad numbering convention used in this paper. (C) Overview of the computational library design procedure. (D–E) Scores for members of three libraries designed to target Bfl-1 (blue), Mcl-1 (green) or Bcl-$x_L$ (red): (D) PSSM$_{SPOT}$ scores, (E) STATIUM z-scores. (F–I) The affinities of library peptides for different Bcl-2 proteins were predicted to be strongly correlated. (F) PSSM$_{SPOT}$ scores for binding to Bcl-$x_L$ versus Bfl-1, (G) PSSM$_{SPOT}$ scores for binding to Mcl-1 versus Bfl-1, (H) STATIUM z-scores for binding to Bcl-$x_L$ versus Bfl-1, (I) STATIUM z-scores for binding to Mcl-1 versus Bfl-1. For (D–I), each point represents one peptide sequence and higher scores correspond to higher predicted affinities for the indicated target. Points on the dashed line have the same low specificity as PUMA BH3 (which is shown with a black open circle).

The following figure supplements are available for figure 1:

**Figure supplement 1.** Affinities of BIM point mutants for different Bcl-2 proteins are predicted to be strongly correlated.

**Figure supplement 2.** Composition of the Bcl-$x_L$, Mcl-1 and Bfl-1-targeted libraries.

**Figure supplement 3.** Scores for members of three libraries designed to target Bfl-1 (blue), Mcl-1 (green) or Bcl-$x_L$ (red).

chemotherapeutic response in vivo (*Ryan et al., 2010*; *Deng et al., 2007*; *Montero et al., 2015*). There has been progress toward creating a panel of reagents specific for each mammalian anti-apoptotic protein that can advance such diagnostic assays. Useful reagents for this purpose include peptides and small molecules that are selective for Mcl-1 (*Foight et al., 2014*; *Kotschy et al. 2016*) or Bcl-x$_L$ (*Dutta et al., 2015*; *Lessene et al., 2013*).

The role of anti-apoptotic protein Bfl-1 in cancer is less characterized than that of Mcl-1 or Bcl-x$_L$, but many lines of evidence suggest that Bfl-1 is also a critical target. In melanoma, Bfl-1 overexpression confers resistance to BRAF inhibitors, and siRNA-mediated knockdown of Bfl-1 induces cell death in melanoma cell lines but not non-malignant cells (*Hind et al., 2015*; *Haq et al., 2013*; *Senft et al., 2012*). Mis-regulation of Bfl-1 is also implicated in hematological malignancies, where elevated levels of Bfl-1 confer resistance to common chemotherapeutic agents. Bfl-1 knockdown suppresses resistance and sensitizes malignant B-cells to chemotherapy (*Brien et al., 2007*). Bfl-1 expression can also counteract the effects of inhibitors of other anti-apoptotic family members (e.g. Mcl-1, Bcl-2) in leukemia and lymphoma (*Fan et al., 2010*). Bfl-1 mRNA is over-expressed in myriad malignancies including solid tumor samples from breast, colon, ovary and prostate tissues (*Beverly and Varmus, 2009*). Thus, Bfl-1 is an intriguing therapeutic target and biomarker for resistance to cytotoxic anticancer drugs.

Identifying Bfl-1-selective interaction inhibitors has proven difficult. Small molecules must compete with an extended protein-peptide interface, and developing small-molecule inhibitors of Bcl-x$_L$, Bcl-2 and Mcl-1 required years of work, guided by intensive NMR studies of fragment binding (*Souers et al., 2013*; *Oltersdorf et al., 2005*; *Kotschy et al., 2016*). Screening has identified small-molecule inhibitors of Bfl-1, but these compounds have IC$_{50}$ values in the high nanomolar to low micromolar range and exhibit only modest specificity for Bfl-1 relative to other Bcl-2 family members (*Mathieu et al., 2014*; *Zhai et al., 2012*; *Cashman et al., 2010*; *Zhai et al., 2008*). Recently, helical bundle proteins that incorporate a BH3 motif have been designed to inhibit Bfl-1 and other anti-apoptotic proteins. These proteins are tight and selective binders, but their function relies on them being folded, and delivering proteins of molecular weight >13 kDa into cells is problematic given current technologies (*Berger et al., 2016*).

An attractive strategy for inhibiting Bcl-2 family proteins is to develop short peptides that mimic the interaction geometry of native Bcl-2 protein complexes (*Figure 1A*). Screening BH3-like peptide libraries previously led to identification of a molecule with ~50 nM affinity for Bfl-1 and 30-fold specificity for Bfl-1 over Mcl-1, (*Dutta et al., 2013*), but this peptide was not shown to induce mitochondrial depolarization in cell-based assays. Identifying Bfl-1 selective peptides is complicated by the extremely large sequence space of short BH3-like helical binders. There are more than 10$^{29}$ possible peptides of length 23 residues. This sequence space is too large to exhaustively search experimentally. Furthermore, the BH3 motif is a weak motif (only three positions are strongly conserved) that does little to restrict possible binders. Another confounding factor is that Bfl-1 interacts with fewer BH3-like peptides than other anti-apoptotic Bcl-2 family paralogs do (*Foight and Keating, 2015*; *DeBartolo et al., 2014*), and no native interaction partners are known to be selective for Bfl-1, suggesting that there may be limited opportunities for achieving specificity.

The results described here showcase our computational/experimental roadmap for designing selective peptide inhibitors. We used computational models to design a focused library of ~10$^7$ candidate binders and screened it to identify three peptides, FS1, FS2 and FS3, that bind tightly and specifically to Bfl-1. Mutational studies and high-resolution structures revealed that the high specificity comes from a BH3 binding mode that is markedly different from what has been seen in prior structures of Bfl-1:BH3 complexes (*Herman et al., 2008*; *Smits et al., 2008*). Importantly, FS1, FS2 and FS3 are specific in BH3 profiling, an assay that tests for MOMP in cells. Subsequent rational introduction of an acrylamide moiety to covalently react with Bfl-1 further enhanced Bfl-1 inhibitor specificity. FS1, FS2, FS3 and their chemical derivatives provide new reagents with utility for studying Bfl-1 biology and a launching point for developing Bfl-1 targeting therapeutics.

## Results

### Computational analysis prioritizes mutations for targeted library design

To reduce the enormous space of possible 23-mer sequences to <$10^7$ candidates that could be tested experimentally, we used computational modeling to design focused combinatorial libraries. We first scored mutations throughout the BH3 motif using: (1) a position-specific scoring matrix (PSSM) derived from SPOT peptide array data (PSSM$_{SPOT}$) and (2) STATIUM, a structure-based statistical potential that previously showed good performance evaluating Bcl-2 protein binding to BH3-like peptides (*Figure 1B,C*) (*DeBartolo et al., 2014*, *2012*). Mutations were modeled in the BIM BH3 motif, with the intention of testing the mutations in the context of both BIM and PUMA BH3 motifs. These two BH3-only proteins, as well as tBID, interact tightly with Bfl-1. BIM and PUMA bind with low-nanomolar affinity to Bfl-1, but also to anti-apoptotic paralogs Bcl-2, Bcl-x$_L$, Mcl-1 and Bcl-w (*Dutta et al., 2013*; *Foight and Keating, 2015*). Thus, our design challenge was to introduce mutations that eliminate off-target binding without destabilizing Bfl-1 binding. Bfl-1 shares 38% binding-groove sequence identity with Mcl-1% and 30% binding-groove identity to Bcl-x$_L$. Bcl-2 and Bcl-w are closely related to Bcl-x$_L$, with 60% sequence identity in the binding groove (*Foight and Keating, 2015*). To model cross-reactivity, we compared how mutations in BIM were predicted to affect binding to Bfl-1 relative to Bcl-x$_L$ and Mcl-1, for which high-quality structures of complexes are available. The predicted binding scores of diverse sequences for the three proteins were highly correlated, and most single mutations were predicted to weaken Bfl-1 binding compared to the wild-type sequence (*Figure 1—figure supplement 1*).

Mutational scoring identified promising positions for introducing sequence variation (helix positions are defined in *Figure 1B* above the sequence of PUMA BH3). Bfl-1, Bcl-x$_L$ and Mcl-1 were predicted to have distinct residue preferences at conserved hydrophobic positions 3d and 4a, consistent with previous observations (*Dutta et al., 2010*). Many mutations at position 4e were predicted to be strongly Bfl-1 selective, which is supported by the observation that peptide binding by both Bcl-x$_L$, and Mcl-1 is weakened by mutations at this position (*Boersma et al., 2008*). Mutations at positions 2a and 3g were also predicted to confer Bfl-1 specificity. In native BH3 motifs, these sites are generally occupied by small charged or polar residues that can form hydrogen bonds/salt-bridges with Bcl-x$_L$ and Mcl-1 groups that are absent in Bfl-1. Finally, the region around sites 2e and 2g has local structural differences in Bfl-1, Mcl-1 and Bcl-x$_L$.

We used in-house software to select degenerate codons at variable sites that optimized the predicted Bfl-1 binding affinity and specificity and that provided chemical diversity in the resulting library (*Dutta et al., 2013*; *Foight et al., 2017*). The final library design included >6.8*$10^6$ unique sequences (*Figure 1—figure supplement 2*), most of which were predicted to be Bfl-1 selective by PSSM$_{SPOT}$ and STATIUM (*Figure 1F–I*). As a control, we designed similarly sized libraries to be selective for Bcl-x$_L$ and Mcl-1 (*Figure 1—figure supplement 2*). PSSM$_{SPOT}$ predicted each library to be enriched in peptides selective for the appropriate target, as shown in *Figure 1D*. In contrast, STATIUM predicted significantly more cross-reactivity for library members (*Figure 1E*, *Figure 1—figure supplement 3*).

### Experimental library screening

Oligonucleotides encoding the peptide libraries designed to be specific for Bfl-1, Bcl-x$_L$ and Mcl-1 were synthesized in the context of BIM and PUMA BH3 sequences. Pooled BIM-based libraries and pooled PUMA-based libraries were then screened separately for tight and selective binding to Bfl-1. Screening the libraries designed for Mcl-1 and Bcl-x$_L$ for binding to Bfl-1, in addition to the library designed to target Bfl-1, provided an opportunity to evaluate the utility of computational library focusing.

We used yeast-surface display to identify selective Bfl-1-binding peptides from our mixed libraries (*Figure 2A*). FACS analysis revealed that the initial libraries had a modest number of cells expressing peptides that bound to Bfl-1 at 100 nM (*Figure 2B*). This is consistent with predictions that less than 6.5% or 4% of the theoretical library would bind as well or better than PUMA, according to PSSM$_{SPOT}$ or STATIUM, respectively.

Most of the peptides that bound Bfl-1 were cross-reactive with one or more other Bcl-2 family proteins (*Figure 2C–G*). This cross-reactivity was expected based on the high correlation of

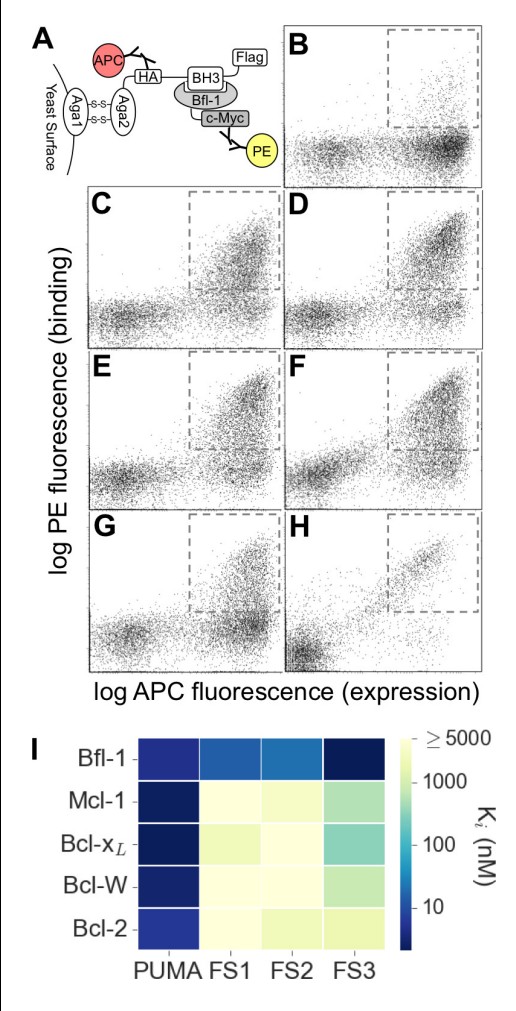

**Figure 2.** Experimental library screening for Bfl-1 affinity and selectivity. (**A**) Yeast-surface display configuration. BH3 peptides were expressed as fusions to Aga2; HA tag expression was detected with APC and Bfl-1 binding was detected with PE. (**B**) FACS analysis showed that only ~5% of cells in the unsorted PUMA libraries bound to Bfl-1 at 100 nM. (**C**) Library binding to 100 nM Bfl-1 after one round of enrichment. (**D–G**) Library binding to off-target proteins (100 nM) after one round of enrichment: (**D**) Bcl-$x_L$, (**E**) Bcl-2, (**F**) Bcl-w, (**G**) Mcl-1. (**H**) Library binding to 100 nM Myc-tagged Bfl-1 in the presence of excess unlabeled competitor (Mcl-1, Bcl-2, Bcl-w and Bcl-$x_L$; 1 µM each) after six rounds of enrichment. (**I**) Inhibition constants determined using fluorescence anisotropy for 23-residue peptides corresponding to PUMA BH3, FS1, FS2 and FS3.

The following source data and figure supplements are available for figure 2:

**Source data 1.** Data collected from competition fluorescence polarization experiments.

*Figure 2 continued on next page*

predicted binding scores for Bfl-1, Mcl-1 and Bcl-$x_L$ and highlights the challenge of identifying specific binders (*Figure 1D–E*, *Figure 1—figure supplement 1*). Six rounds of positive, negative and/or competition FACS screening were used to isolate cells that expressed the tightest and most Bfl-1-selective peptides (*Figure 2—figure supplement 1*). Mcl-1, Bcl-$x_L$, Bcl-2 and Bcl-w were included in the screen as untagged competitors. Early screening provided many Bfl-1 selective hits from the PUMA libraries, but few from the BIM libraries, so the BIM libraries were not pursued (*Figure 2—figure supplement 2*). After several rounds of competition screening, the PUMA library was enriched in cells displaying peptides that bound to Bfl-1 at 100 nM in the presence of 40-fold excess unlabeled competitor (*Figure 2H*).

Fifty colonies isolated in the final round of screening were sequenced, providing 13 unique sequences: nine sequences were from the Bfl-1 specific library, two were from the Bcl-$x_L$ library, and two were from the Mcl-1 library (*Figure 2—figure supplement 3*). We tested three Bfl-1 selective peptides that were recovered two or more times (FS1, FS2 and FS3). FS1, FS2 and FS3 were all derived from the Bfl-1 targeted library, although FS1 also contained one mutation caused by a spurious single-base pair mutation. FS1, FS2 and FS3 each had reduced affinity for Bfl-1 relative to PUMA, but significantly increased specificity (*Figure 2I* and *Figure 2—figure supplement 4*). FS1 bound Bfl-1 with $K_i$ = 15 nM and at least 150-fold specificity for Bfl-1 relative to Bcl-$x_L$, Bcl-2, Bcl-w and Mcl-1.

To analyze enrichment trends and to assess the success of our library design, we deep sequenced samples from the naïve pool and from pools collected after 3, 4, 5 and 6 rounds of sorting (sorting conditions are detailed in *Figure 2—figure supplement 1*). The naive pool was diverse and not dominated by any particular subset of sequences. In contrast, FS1 (38% of sequences, the most prevalent library member), FS2 (25% of sequences), and many other peptides from the Bfl-1 targeted library were prominent in the final screening pool. Analysis of sequential pools showed that peptides from the Bfl-1 targeted library were substantially enriched relative to peptides from the Bcl-$x_L$ and Mcl-1 targeted libraries (*Figure 3A*). Of the unique sequences in the final pool, 73.9% were from the Bfl-1 targeted library (*Figure 3B*).

We scored peptides from the Bfl-1 targeted library that passed all rounds of screening with the STATIUM and PSSM$_{SPOT}$ models used in library design (*Figure 3C–F*). Most sequences

*Figure 2 continued*

**Figure supplement 1.** The PUMA BH3 library was screened to enrich for selective binders of Bfl-1.

**Figure supplement 2.** FACS analysis of the designed libraries after first two rounds of sorting (FL2, see *Figure 2—figure supplement 1*).

**Figure supplement 3.** Conventionally sequenced clones from pool FL6'.

**Figure supplement 4.** Peptide affinities for Bfl-1, Bcl-$x_L$, Mcl-1, Bcl-2 and Bcl-w.

**Figure supplement 5.** FS2 mutations made in a BIM background generate a weak binder of Bfl-1.

were predicted to have improved selectivity for Bfl-1 relative to PUMA (98–99% with improved specificity over Bcl-$x_L$ or Mcl-1 by PSSM$_{SPOT}$, and 95% or 62% with improved specificity over Bcl-$x_L$ or Mcl-1, respectively, by STATIUM). The selected sequences were not among those predicted by either model to be the tightest or most Bfl-1 selective in the theoretical library.

## The binding mode of Bfl-1-selective peptides

FS1, FS2 and FS3 included mutations to larger residues than those in PUMA at their N-termini (red in *Figure 4B,C*), and smaller residues at their C-termini (blue in *Figure 4B,C*). Deep sequencing of additional selective sequences supported this trend: Of 612 unique peptide sequences from the final round of sorting that originated from the Bfl-1 targeted library sequences, 364 showed this type of residue size patterning at the same sites (sequence logo in *Figure 4A*).

To assess whether the combination of large and small residues played a role in establishing binding specificity, we tested PUMA/FS2 chimeric peptides for binding to all five anti-apoptotic proteins. Mutating PUMA to introduce smaller residues at positions 2g, 3d, 4a and 4e differentially impaired binding to all receptors and resulted in weak yet specific binding to Bfl-1 (*Figure 4—figure supplement 1*). Mutating residues at the N-terminus of PUMA to larger residues at positions 2a and 2e gave a modest 2.3-fold increase in affinity for Bfl-1. But the same mutations in the context of smaller residues at positions 2g, 3d, 4a and 4e improved affinity for Bfl-1 by 28.6-fold (*Figure 4D*). The different effects of these mutations, when made in different contexts, indicates an energetic coupling consistent with a structural repositioning of the designed peptides in the groove of Bfl-1.

To better understand the structural basis for the epistasis, we solved X-ray crystal structures of Bfl-1 bound to PUMA, at 1.33 Å resolution, and of Bfl-1 bound to FS2 at 1.2 Å resolution (*Supplementary file 1*). In comparison with all available X-ray structures of BH3 peptides bound to human or murine Bfl-1, PUMA and FS2 each adopt new, distinct positions in the binding groove (*Figure 5A* and *Figure 5—figure supplement 1*). FS2 is shifted 1.2 Å and rotated 17° in the binding groove compared to its parent peptide PUMA. The peptide C-terminus, which harbors the large-to-small mutations, is repositioned more dramatically than the N-terminus (*Figure 5B*). Despite the shifts in peptide binding geometry, the structures of Bfl-1 in these newly solved complexes are highly similar. The all-atom RMSD for residues in the binding pocket (within 5 Å of the BH3 peptide) of Bfl-1:FS2 vs. Bfl-1:PUMA is <0.7 Å and is 1.05 Å for Bfl-1:FS2 vs. Bfl-1:BIM (*Herman et al., 2008*).

Further structural analysis showed that the Bfl-1:FS2 complex supports several key side-chain interactions that are absent in Bfl-1:PUMA and that may be important for selective binding. Surprisingly, aspartate at position 3f (D3f) in FS2, which is strongly conserved in known BH3 motifs, makes different interactions than what is observed in numerous previously solved Bcl-2 complex structures. D3f typically forms a salt bridge with arginine 88 (R88) in helix four in Bfl-1 or the corresponding arginine in Bcl-$x_L$, Mcl-1, Bcl-w or Bcl-2 (*Figure 5C*). In the Bfl-1:FS2 structure, the carboxylate of D3f is shifted 5.6 Å away from the guanidinium group of R88, and is highly solvent exposed (*Figure 5C*). Because D3f does not form the canonical D3f:R88 interaction and is solvent exposed, we reasoned that FS2 should tolerate mutations at this site. This was confirmed by the tight binding of six peptides with alanine, serine, asparagine, glutamate, histidine or tyrosine at this position (*Figure 5—figure supplement 2*). Disruption of the D3f:R88 salt bridge would be expected to reduce affinity for Bfl-1 and for all of the other anti-apoptotic receptors. However, in the Bfl-1:FS2 complex this change may be partially compensated by hydrogen bonding on the opposite side of the FS2 helix between arginine at position 3c (R3c) of FS2 and asparagine 51 (N51) of Bfl-1 (*Figure 5E*). In Bfl-1:FS2, position 3c is positioned closer to helix 2 of Bfl-1 than in Bfl-1:PUMA, allowing R3c to fill the space left by an adjacent methionine-to-alanine mutation at 3d when it adopts this hydrogen-bonded position.

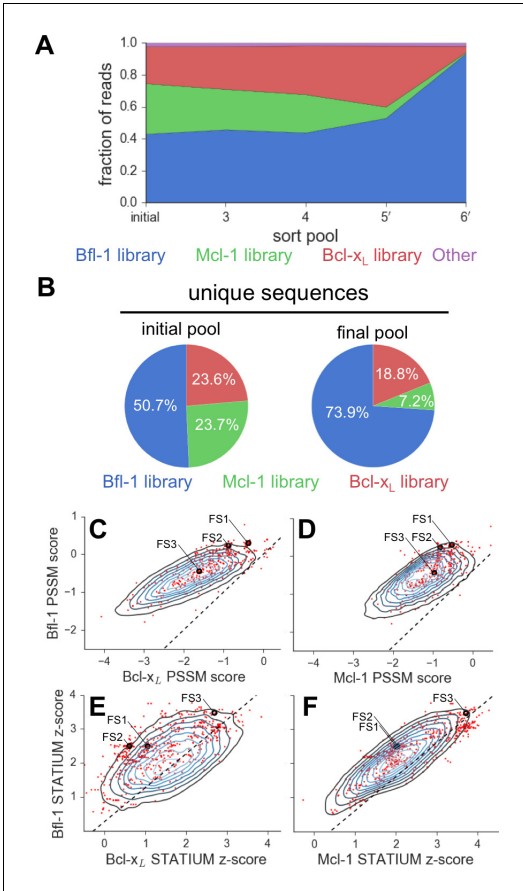

**Figure 3.** Evaluation of the library design. (**A**) Sequences from the Bfl-1 library were preferentially enriched during sorting. Sequences with no more than one amino-acid mutation from the Bfl-1 (blue), Mcl-1 (green), or Bcl-x$_L$ (red) targeted libraries are plotted. Other sequences are shown in magenta. (**B**) The large majority of unique sequences in the final pool originated from the Bfl-1 library (colors as in part A). (**C–F**) Comparison of PSSM$_{SPOT}$ and STATIUM scores for the library before and after sorting. Peptides from the final sorted pool (red dots) are superimposed on the distribution of scores for the theoretical library (blue contour plots). Points to the left of the dotted lines correspond to peptides predicted to bind more selectively to Bfl-1 than does PUMA, with respect to the indicated competitor protein (Bcl-x$_L$ in C and E, Mcl-1 in D and F). Scores for FS1, FS2 and FS3 are indicated.

The following source data is available for figure 3:

**Source code 1.** Deep sequencing data analysis.

N51 at this position of helix two is unique to Bfl-1 among the human anti-apoptotic proteins (*Figure 5—figure supplement 3*).

Other structural differences between PUMA and FS2 binding are apparent near the

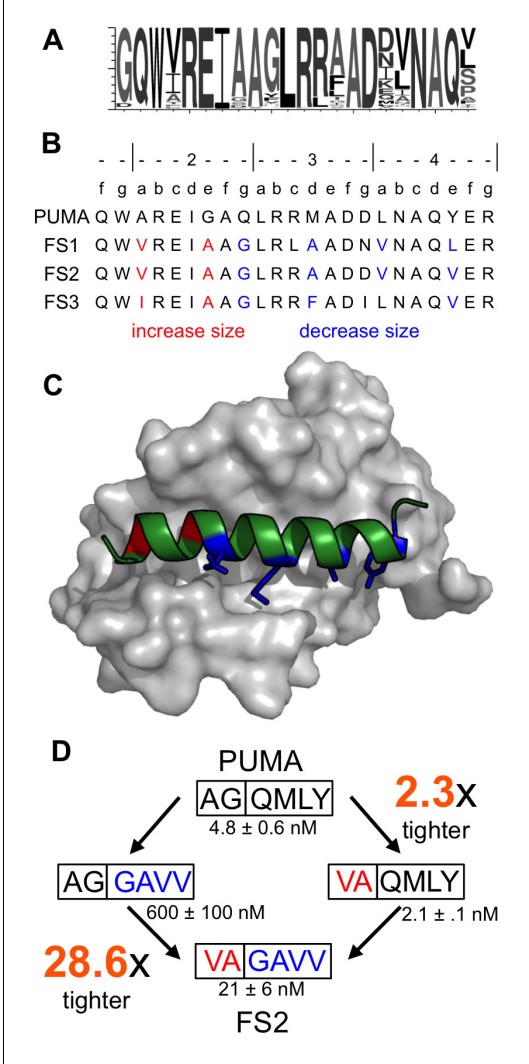

**Figure 4.** Epistatic mutations in PUMA confer Bfl-1 binding specificity. (**A**) Sequence logo of unique peptide sequences in the final sorted pool from the Bfl-1 targeted library. (**B**) Location of mutated sites in FS1, FS2 and FS3. Mutations at positions 2a and 2e are in red and positions 2g, 3d, 4a and 4e are in blue. (**C**) Structure of Bfl-1 (gray surface) bound to PUMA (green, this work) with residues at positions 2a and 2e in red and those at 2g, 3d, 4a and 4e in blue. (**D**) Non-additive mutational energies for PUMA/FS2 chimeric proteins indicate coupling between N- and C-terminal mutations. Data are K$_i$ ± SD of three or more independent fluorescence anisotropy competition experiments.

The following source data and figure supplement are available for figure 4:

**Source data 1.** Data collected from competition fluorescence polarization experiments.

**Figure supplement 1.** Affinities of FS2 chimeric proteins binding to Bfl-1, Bcl-x$_L$, Mcl-1, Bcl-2 and Bcl-w.

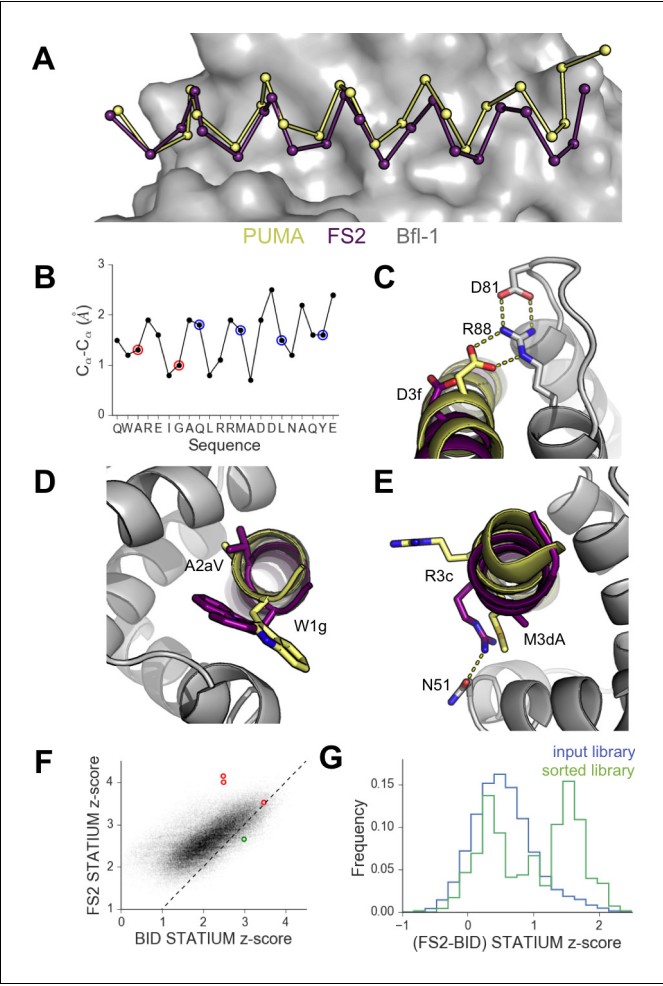

**Figure 5.** High-resolution structures of PUMA and FS2 bound to human Bfl-1. (**A**) Binding groove of Bfl-1 (gray, surface) with PUMA (yellow) and FS2 (purple). (**B**) $C_\alpha$- $C_\alpha$ shifts between FS2 and PUMA. Sites with larger/smaller residues in FS2 are indicated in red/blue. (**C**) The canonical Bfl-1:BH3 salt bridge between D3f and R88 is observed in the Bfl-1:PUMA complex but not the Bfl-1:FS2 complex. (**D**) Tryptophan at 1g is rotated into the Bfl-1 binding groove in the Bfl-1:FS2 complex and away from the binding groove in the Bfl-1:PUMA complex. (**E**) In contrast with the solvent exposed arginine at position 3c of the Bfl-1:PUMA complex, R3c is oriented into the BH3 binding groove in the Bfl-1:FS2 complex, forming a hydrogen bond with N51 of Bfl-1. (**F**) Bfl-1 targeted library sequences score better on the Bfl-1:FS2 structure than on the Bfl-1:BID structure used for the initial library design; higher scores predict tighter binding. STATIUM z-scores for the Bfl-1 targeted library are in blue. FS1, FS2 and FS3 are indicated in red and PUMA in green. (**G**) Sorting enriched sequences that score better on the Bfl-1:FS2 template than on the Bfl-1:BID template. STATIUM z-scores for the input Bfl-1 library are in blue and scores for sequences identified after the final round of screening are in green.

The following figure supplements are available for figure 5:

**Figure supplement 1.** Comparison of PUMA and FS2 binding geometry with that in other crystal structures of BH3:Bfl-1 complexes deposited in the PDB.

**Figure supplement 2.** FACS analysis of cells displaying FS2 or FS2 with single point mutants at position 3f.

**Figure supplement 3.** Multiple-sequence alignment of helices 2–4 of human anti-apoptotic Bfl-1 homologs.

**Figure supplement 4.** Residues in FS2 are not readily accommodated in the PUMA -binding geometry.

N-terminal end of the peptide. Modeling FS2 mutations in the Bfl-1:PUMA structure suggested that the small-to-large mutation of alanine at position 2a in PUMA to the valine in FS2 would result in steric clashes with helix 4 of Bfl-1 for all backbone-dependent rotamers (*Figure 5—figure supplement 4*). This change is accommodated by the shift in the Bfl-1:FS2 structure. Also, a rotation of FS2 in the Bfl-1 binding groove partially buries the phenylalanine at position 1g that is solvent-exposed in the PUMA complex, which may be energetically favorable (*Figure 5D*).

Because the altered binding mode of FS2 is expected to impact predictions made using structure-based models, we re-scored the designed Bfl-1 library on the shifted Bfl-1:FS2 structure using STATIUM. FS1 and FS2 scored much better (higher) on the shifted model than on the original model, whereas PUMA scored better on the original model (*Figure 5F*). Analysis of the entire pool of sequences that passed screening showed that these peptides were enriched in sequences that scored better on the shifted model, compared to the input library, consistent with our observation of size patterning in the majority of these sequences (*Figure 5G*).

## Structural analysis of off-target binding to Mcl-1

To better understand the structural basis of FS2-binding specificity, we solved the X-ray crystal structure of FS2 bound to Mcl-1 at 2.35 Å resolution. FS2 binding to Mcl-1 is >100 fold weaker than binding to Bfl-1. Similar to the way FS2 binds to Bfl-1, FS2 engages Mcl-1 in a shifted orientation relative to BIM (*Figure 6A,B*). As is the case for FS2 binding to Bfl-1, this shift re-positions the highly conserved aspartate at peptide position 3f to a location 4.8 Å away from Mcl-1, disrupting the canonical salt bridge with arginine 92 (*Figure 6C*). This disruption would be expected to reduce affinity for Mcl-1, but it doesn't account for the specificity of FS2 for Bfl-1, because the salt bridge is lost in both complexes. There are other differences between the Bfl-1:FS2 and Mcl-1:FS2 structures that may account for some of the affinity difference. For example, R3c in FS2 forms a hydrogen bond with N51 of Bfl-1, but does not form an equivalent interaction with Mcl-1 and is instead solvent exposed (*Figure 6D*). In Mcl-1, there is an alanine (A55) at this site, and an adjacent histidine (H53) would be expected to clash with R3c if it adopted this conformation. The N-terminus of FS2 is also buried further into the binding groove of Bfl-1 than Mcl-1 (*Figure 6E*). The Bfl-1 binding groove is wider in this region than the Mcl-1 binding groove, as illustrated by aligning many Bfl-1 and Mcl-1 structures (*Figure 6—figure supplement 1*). This region of the groove is formed by helices 3 and 4. There is an amino acid insertion in the loop between helices 3 and 4 that is unique to Bfl-1 that likely contributes to the distinct structural environment of Bfl-1 in this region (*Figure 5—figure supplement 3*).

## Biological activity of designed Bfl-1 inhibitors

We tested our designed peptides for Bfl-1 selective targeting by carrying out BH3 profiling of cells with known dependencies on anti-apoptotic proteins. In this assay, peptides are titrated into permeabilized cells, and mitochondrial depolarization is measured using the voltage-sensitive dye JC-1 (*Figure 7A*) (*Deng et al., 2007*). We tested the apoptotic sensitivity of BCR-ABL-expressing B-lineage acute lymphoblastic leukemia cell lines engineered to depend on Bcl-2, Bcl-$x_L$, Mcl-1 or Bfl-1 overexpression for survival (*Koss et al., 2016*). The percent depolarization from these assays is shown in *Figure 7B*. In comparison with a shorter, truncated PUMA BH3 (PUMA1e-4c, PUMA$_{sh}$), which promoted mitochondrial depolarization in all of the cell lines tested, at 100 nM the Bfl-1 selective inhibitors FS1, FS2 and FS3 promoted depolarization only in Bfl-1 dependent cells. An inactive PUMA$_{sh}$ mutant, PUMA L3aA;D3fA (PUMA 2A) was used as a negative control (*Ryan and Letai, 2013*). EC$_{50}$ values for inducing mitochondrial permeabilization in the engineered cell lines agreed well with trends in Bfl-1-binding affinities, as expected based on the mechanism of action (*Figure 7C*).

As an additional test for on-pathway activity, we measured cytochrome c release in the same engineered cell lines in response to peptide treatment, using iBH3 profiling (*Ryan et al., 2016*). The specificity pattern observed when monitoring cytochrome c release was consistent with that obtained by BH3 profiling read out using JC-1 (*Figure 7D*, *Figure 7—figure supplement 1*). A Mcl-1 selective peptide, MS1 was used as a control (*Foight et al., 2014*). In both assays, FS3 promoted mitochondrial depolarization more potently than FS1 or FS2, but was less selective, with significant cross reactivity at 30 μM peptide concentration.

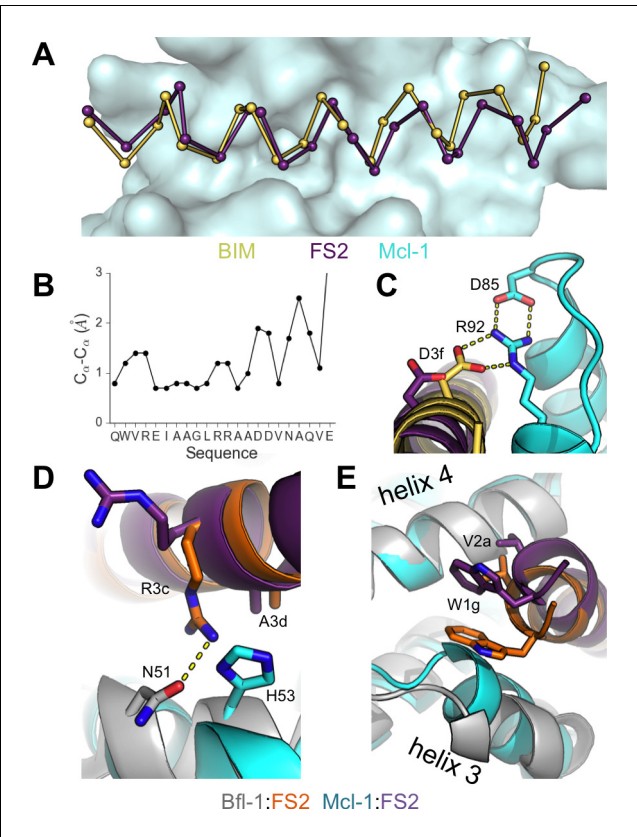

**Figure 6.** Crystal structure of FS2 bound to human Mcl-1. (**A**) Binding groove of Mcl-1 (blue, surface) with BIM (yellow, 2PQK [*Fire et al., 2010*]) and FS2 (purple). (**B**) $C_\alpha$- $C_\alpha$ shifts between FS2 and BIM when bound to Mcl-1. (**C**) The canonical Bfl-1:BH3 salt bridge between D3f and R92, formed in Mcl-1:BIM, is not observed in the Mcl-1: FS2 complex. (**D**) In contrast with the arginine at position 3c of the Bfl-1:FS2 complex, which makes packing and hydrogen-bond interactions the interface, R3c is oriented away from the BH3 binding groove in the Mcl-1:FS2 complex. (**E**) The Mcl-1 binding groove between helix 3 and helix four is narrower than the Bfl-1 binding groove, and the N-terminus of FS2 is shifted in the Mcl-1:FS2 structure in comparison with the Bfl-1:FS2 complex.

The following figure supplement is available for figure 6:

**Figure supplement 1.** Alignment of all crystal structures in the PDB of Bfl-1/Mcl-1 bound to BH3 peptides.

FS1, FS2 and FS3 were based on the sequence of PUMA BH3, which has been proposed to directly activate apoptosis through interactions with BAK and BAX (*Dai et al., 2014*; *Edwards et al., 2013*). To test the possibility that FS1, FS2 or FS3 may directly activate BAK and BAX, we measured cytochrome c release in two 'unprimed' cell lines (PC-3 and SF295). Unprimed cells require BAK/ BAX activators to release cytochrome c (*Certo et al., 2006*). We observed cytochrome c release in cells treated with BIM or PUMA BH3 but not in cells treated with as much as 100 µM FS1 or FS2 (*Figure 7—figure supplement 2*). Treatment with FS3 or PUMA$_{sh}$ peptide only led to cytochrome c release at high concentrations (~30-100 µM) (*Figure 7—figure supplement 2*). This may indicate that FS3 and PUMA$_{sh}$ have very weak activator function. Taken together, our data show that FS1 and FS2 are not themselves activators, but that they instead act as apoptotic sensitizers by competing with activators or with BAX or BAK for binding to anti-apoptotic proteins, as intended in our design scheme.

## Covalent inhibitors of Bfl-1 enhance specificity

Our initial sorts for Bfl-1 selective binders identified many sequences that included cysteine at position 1g or 2b (*Figure 8—figure supplement 1*). Interestingly, cysteines encoded at several other

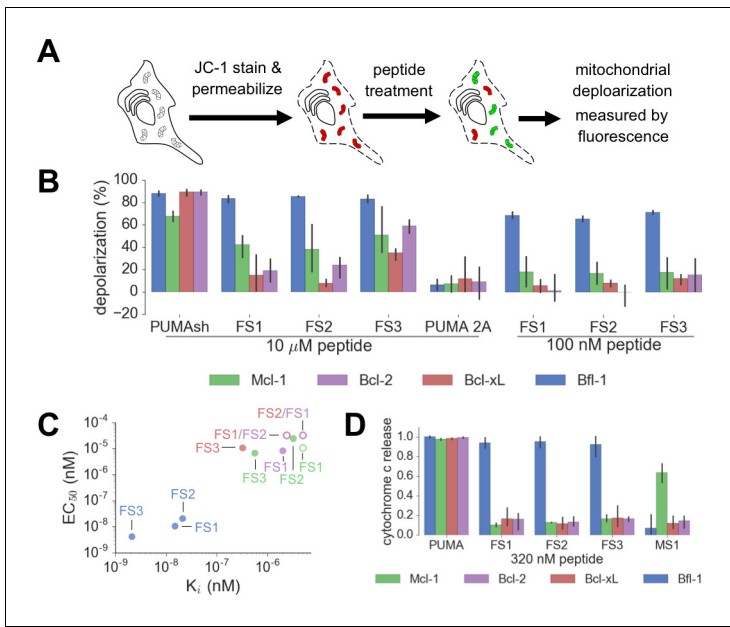

**Figure 7.** Designed Bfl-1 inhibitors selectively induce MOMP in Bfl-1 dependent cells. (**A**) The BH3 profiling assay detects MOMP by monitoring JC-1 fluorescence in permeabilized cells treated with different peptides. (**B–C**) Depolarization of mitochondria induced by designed peptides in four cell lines that depend on ectopic expression of Mcl-1, Bcl-2, Bcl-$x_L$ or Bfl-1 for survival. (**C**) Correlation between $K_i$ in solution studies and $EC_{50}$ values in BH3 profiling. Open circles indicate lower bound estimates of $EC_{50}$ or $K_i$. (**D**) Cytochrome c release from the same cell lines in **B** and **C**. Data are mean ± SD of three or more independent measurements.

The following source data and figure supplements are available for figure 7:

**Source data 1.** Data collected from BH3 profiling and cytochrome c release assays.

**Figure supplement 1.** At low concentrations, FS1, FS2 and FS3 selectively induced cytochrome c release only in Bfl-1 dependent cell lines. iBH3 was performed on highly primed cells of known anti-apoptotic protein dependency.

**Figure supplement 2.** iBH3 was performed on unprimed cells to test for activation function.

positions along the BH3 motif were not enriched. Furthermore, cysteine was not enriched in previous screens for Bfl-1 binding (**Dutta et al., 2013**). This observation led us to hypothesize that Bfl-1 binding selectivity could be improved in non-reducing conditions if the peptide ligand formed a disulfide bond with cysteine 55 (C55) of Bfl-1, which is adjacent to the binding cleft of Bfl-1 and unique to Bfl-1 among Bcl-2 family paralogs (**Figure 8A**). Testing yeast-displayed peptides for binding to a Bfl-1 cysteine-to-serine (C55S) mutant confirmed that PUMA and BIM bound to Bfl-1 C55S, whereas the majority of the peptides in the cysteine-enriched pool bound to wild-type Bfl-1 but not to Bfl-1 C55S (**Figure 8—figure supplement 2**). Rescreening our library using Bfl-1 C55S led to the identification of FS1, FS2 and FS3, as described above. But, in addition, the discovery that BH3 peptides in our library could access a unique, reactive cysteine in Bfl-1 led us to design covalent inhibitors based on these peptides.

We used structure-based modeling to choose appropriate cysteine-reactive electrophiles and optimize their placement in different BH3 positions in the 2VM6 structure of Bfl-1 bound to BIM BH3 (**Herman et al., 2008**). Our two most promising designs featured N-terminal Michael acceptors at position 1g (FS2_1gX; **Figure 8B**) or 1f (FS2_1fX; **Figure 8C**) of peptide FS2. We tested our designs for covalent modification of Bfl-1 and Bfl-1 C55S using gel-shift assays. Both FS2_1gX and FS2_1fX modified Bfl-1 once or less when applied at micromolar concentrations, whereas Bfl-1 C55S (which contains two other solvent-exposed cysteine residues) did not react with these electrophilic peptides

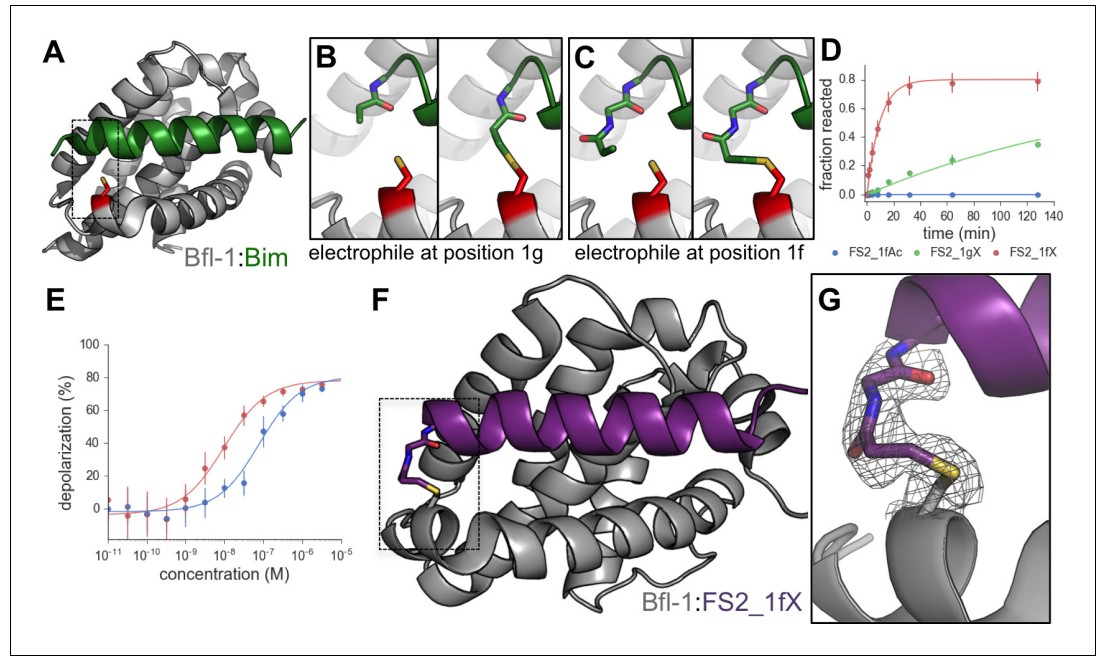

**Figure 8.** An electrophilic variant of FS2 reacts covalently with Bfl-1. (**A**) C55 in Bfl-1 is close to the BH3 binding groove in BIM:Bfl-1 structure 2VM6 (*Herman et al., 2008*). (**B–C**) Modeling suggested two ways in which an N-terminal acrylamide group could be incorporated into a BH3 peptide with good reaction geometry, leading to peptides FS2_1gX (modification shown in B) or FS2_1fX (modification shown in C) (**D**) FS2_1fX (red) reacted more rapidly with Bfl-1 than FS2_1gX (green). Bfl-1 crosslinking as a function of reaction time was measured using gel-shift assays; data are mean ± SD of two or more independent measurements. Crosslinking did not occur with the acetylated control peptide FS2_1fAc (blue). (**E**) FS2_1fX (red) was more potent than FS2_1fAc (blue) in BH3 profiling assays of Bfl-1 dependent cells. Data are mean ± SD of three or more independent measurements. (**F**) X-ray structure of Bfl-1 covalently cross-linked to FS2_1fX. (**G**) Electron density map of covalent crosslink between FS2_1fX and Bfl-1.

The following source data and figure supplements are available for figure 8:

**Source data 1.** Data collected from gel shift assays.
**Figure supplement 1.** Conventionally sequenced clones from pool FL6 and their frequencies.
**Figure supplement 2.** Library members bind covalently to Bfl-1 cysteine 55.
**Figure supplement 3.** Kinetics of the reaction of Bfl-1 with electrophilic peptides.
**Figure supplement 4.** Depolarization of mitochondria induced by designed peptides, including covalent inhibitor FS2_1gX, in four cell lines that depend on ectopic expression of Mcl-1, Bcl-2, Bcl-$x_L$ or Bfl-1 for survival.

for at least 6 hr (*Figure 8—figure supplement 3*). Using densitometry, we measured the fraction of Bfl-1 reacted as a function of time for both designs. FS2_1fX reacted with Bfl-1 with a half-life of 6.5 min and FS2_1gX reacted more slowly with a half-life of 138 min (*Figure 8D*). We tested FS2_1fX in BH3 profiling and found that it improved on-pathway targeting of Bfl-1 compared to N-terminally acetylated control (*Figure 8E*) and was selective for Bfl-1 (*Figure 8—figure supplement 4*). We solved a crystal structure of FS2_1fX bound to Bfl-1 that showed clear electron density consistent with a covalent bond to C55, as designed (*Figure 8F,G*).

# Discussion

Bfl-1 is implicated in cancer progression, and inhibiting its anti-apoptotic function may be therapeutically beneficial. Because the role of Bfl-1 in disease is less characterized than that of Mcl-1 or Bcl-2, selective targeting agents will likely be critical for disentangling the roles of different anti-apoptotic proteins. Prior efforts to identify Bfl-1 selective inhibitors provided molecules with only modest binding affinity and/or selectivity (*Mathieu et al., 2014*; *Zhai et al., 2012*; *Cashman et al., 2010*; *Zhai et al., 2008*; *Dutta et al., 2013*). More recent work has shown, as we demonstrate here, that reaction of an electrophilic group on a Bfl-1 inhibitor with a cysteine near the BH3-binding site confers both infinitely tight and covalently selective interactions with Bfl-1 in preference to other Bcl-2 protein family members. This strategy has been used to target Bfl-1 with a BIM peptide chemically modified to react with cysteine (*Huhn et al., 2016*; *Barile et al., 2017*; *de Araujo et al., 2017*). But BIM is a promiscuous binder of Bcl-2, Bcl-x$_L$, Bcl-w and Mcl-1, as well as an activator of BAX and BAK (*Letai et al., 2002*). Thus, although Bfl-1 is the only covalent target of electrophilic BIM, this approach does not provide an optimal strategy for dissecting the contributions of Bfl-1 to cell survival, because BIM peptides are expected to have many effects on the apoptotic protein-interaction network. Here, we addressed this by successfully re-engineering PUMA, a promiscuous binder of anti-apoptotic proteins, to make it highly selective for Bfl-1 both as a non-covalent and as a covalent inhibitor.

In re-designing PUMA, we confronted the enormous space of possible mutational variants. The challenge was to identify combinations of mutations that would reduce binding to Bcl-2, Bcl-x$_L$, Bcl-w and Mcl-1, while not substantially weakening Bfl-1 interaction. We used computational modeling and existing experimental data to guide our design of focused libraries of peptides predicted to provide the desired specificity. We tested libraries of combinations of the best-ranked mutations both in the context of BIM and in the context of PUMA.

Interestingly, our experimental results showed stark differences in binding behavior between two libraries that introduced the same mutations into BIM vs. PUMA BH3 peptides. Our carefully designed library was rich in Bfl-1 selective binders when tested in a PUMA background, but poor in binders when encoded in a BIM BH3 sequence. This was not because PUMA is a tighter or more selective binder of Bfl-1 than is PUMA; published data indicate that BIM BH3 binds at least as tightly to Bfl-1 as does PUMA BH3 (*Dutta et al., 2013*). Also, deep sequencing of the two input libraries indicated that they were of similar quality; each had only a small percentage of sequences that were not designed (6.6% for the BIM library and 4.4% for the PUMA library). Given that dramatic differences were observed between the libraries early in screening (pool FL2, *Figure 2—figure supplement 2*), and that the PUMA library still contained just 5.9% non-designed mutations in pool FL3, it is unlikely that random mutations were the source of the observed differences.

We think it likely that the difference between the PUMA vs. BIM libraries arose because background sequence differently influenced the contributions of library mutations, or combinations of mutations, to binding. For example, this could be the case if PUMA variants but not BIM variants could undergo the conformational shifts we observed in our structures. Consistent with this, we found that introducing the six mutations of FS2 into BIM BH3 rather than PUMA BH3, giving BIM-FS2, weakened BIM BH3 binding to Bfl-1 by >1000 fold; *Figure 2—figure supplement 5*. Differential effects of mutations based on BH3 context were also observed by DeBartolo et al., who reported modest correlation between the influence of point mutations on Mcl-1 binding tested in the context of BIM vs. NOXA BH3 (Pearson r ~ 0.55), and between the effect on Bcl-x$_L$ binding of mutations made in BIM vs. BAD (r ~ 0.78) (*DeBartolo et al., 2012*). Our results are also consistent with the experiments of Dutta et al., who studied Bfl-1 binding selectivity in BIM-based libraries but identified only modestly selective peptides in screening (*Dutta et al., 2013*). Other groups have reported similarly dramatic background effects of context in protein engineering, for example in antibody libraries, and have obtained different success rates from screens and selections that started with different framework sequences (*Boyer et al., 2016*).

Epistasis between mutations contributed to selective Bfl-1 binding (*Figure 4D*). Crystal structures of Bfl-1 bound to PUMA vs. FS2 show that epistasis arises from a substantial alteration of the Bfl-1-binding mode. FS2 binds in a shifted and rotated orientation relative to PUMA (*Figure 5A*), and does not make a key salt bridge that is conserved in nearly all structures of Bcl-2 protein complexes. Interestingly, sorting for Bfl-1 selective binders enriched sequences that score better when modeled

using the FS2 binding conformation than with the BIM or PUMA-binding geometry (*Figure 5G*), suggesting that this structural shift may be a common feature of many sequences that we identified and a general strategy for achieving Bfl-1 specificity. The FS2 structures now provide a springboard to permit the design of further improvements in Bfl-1 selectivity, affinity, cell permeability, and other physicochemical properties. For example, the structures of FS2 bound to Bfl-1 can be used to design chemical crosslinks to reinforce helicity and promote cellular uptake (*Rezaei Araghi and Keating, 2016*; *Rezaei Araghi et al., 2016*; *Bird et al., 2016*; *Walensky et al., 2004*), the structure of FS2 bound to Mcl-1 can be used to further reduce Mcl-1 binding, and the structure of FS2_1fX can be used as a platform for the design of therapeutic peptides and small molecules that covalently target Bfl-1.

By combining the strengths of computational design and library screening, we successfully identified rare peptides with desired binding specificities. It is unlikely that computational approaches alone could have identified FS1, FS2 or FS3 in the absence of structural templates for the FS2-binding mode. Also, library sequences that were predicted to give the best Bfl-1 affinity and specificity over Bcl-x$_L$ and Mcl-1 (the top/left edge of the score distribution for sequences in *Figure 1F–I*) were not among the hits recovered in screening. This could be because of deficiencies in the models, or because the experiments included competitor homologs Bcl-w and Bcl-2, which were not modeled in library design. A structure of Bcl-w bound to a BH3 peptide was not available when we deigned our libraries, so designing specificity against Bcl-w would have been subject to the inaccuracies of homology models. It is also possible that even the highest affinity sequences that bind in the geometry that we modeled cannot match the tight binding or specificity that can be achieved with a conformational shift.

Conversely, unguided library approaches (including random mutagenesis) would probably not generate FS1, FS2 or FS3. More than $6 \times 10^{12}$ sequences are six mutations away from 23-residue PUMA BH3, and most mutations are predicted to weaken binding. Furthermore, our models predict that most mutations will have correlated effects on binding to different Bcl-2 family members (*Figure 1—figure supplement 1*), which dramatically reduces the random chance of finding mutations that confer specificity. Library approaches that take advantage of iterative randomization would have difficulty finding sequences that contain synergistic mutations like those in FS2.

Random mutagenesis is a powerful tool for exploring a local sequence space (e.g. by error-prone PCR), and this could be a strategy for further improving peptides identified in library screens such as this one. The low frequency of non-designed mutations in our libraries introduced a random sampling element, but this did not appear to be important for success in this study. FS2 and FS3 were included in the designed library, and FS1 had only a single non-designed mutation, at a solvent-exposed site.

The peptides we designed in this work have immediate value as biochemical reagents and tools for therapeutic development. FS1, FS2 and FS3 selectively trigger apoptosis of Bfl-1-dependent cells in a BH3 profiling assay, and given that Bcl-2, Mcl-1 and Bcl-x$_L$ dependencies are predictive of therapeutic response to cytotoxic anticancer drugs (*Ryan et al., 2010*; *Deng et al., 2007*; *Montero et al., 2015*), we speculate that diagnosing Bfl-1 dependence using these peptides will provide additional predictive power to guide the use of existing treatments (*Ni Chonghaile et al., 2011*). Furthermore, recent studies with an electrophilic variant of BIM show that targeting Bfl-1 enhances cytotoxicity and caspase 3/7 activity in at least 3 Bfl-1 expressing melanoma cell lines (*Huhn et al., 2016*). Finally, the high affinity and selectivity of these peptides, along with the available structural data, make them promising starting points for the development of peptide- or small-molecule therapeutics directly targeting Bfl-1. Therapeutic applications using peptides require a solution to the cell delivery problem. But chemical modification by hydrocarbon stapling, carefully optimized, has proven effective for delivering other helical BH3 peptides into cells (*Rezaei Araghi and Keating, 2016*; *Rezaei Araghi et al., 2016*; *Bird et al., 2016*; *Walensky et al., 2004*). We are optimistic that similar modifications will be effective for FS1, FS2 and/or FS3, and we predict rapid development of a range of cell-deliverable Bfl-1 targeting agents that draws on lessons learned from Bcl-x$_L$ and Mcl-1 inhibitor design as well as the new structural insights that we provide here.

## Materials and methods

### Peptide synthesis and purification

Library peptides, the PUMA BH3 peptide and PUMA BH3 peptide mutants had N-terminal acetlyation and C-terminal amidation. Fluoresceinated BIM (fluorescein-IWIAQELRRIGDEFNAYY) BH3 was synthesized with N-terminal 5/6-fluorescein amidite and C-terminal amidation. Covalent peptide inhibitors had N-terminal acrylamide and C-terminal amidation. Peptides were synthesized by the MIT Biopolymers Laboratory. The crude synthesis product was purified by HPLC on a C18 column with a linear gradient of acetonitrile in water. Peptides were verified by mass spectrometry.

### Fluorescence polarization assay

Competition fluorescence polarization experiments were performed by titrating 0–10 µM of unlabeled peptide into 50 nM receptor plus 25 nM fluoresceinated BIM (fluorescein-IWIAQELRRIGDEF NAYY) in FP buffer (25 mM Tris pH 7.8, 50 mM NaCl, 1 mM EDTA, 5% DMSO, 0.001% v/v Triton-X). C-myc-tagged receptors were used for all Bcl-2 homologs, as previously described (*DeBartolo et al., 2012*; *Dutta et al., 2010*). Plates were mixed at 23°C for 3 hr. Plates were read again at 24 to check equilibration. Experiments were done in at least triplicate. Data were fit, as described for competition fluorescence anisotropy experiments in *Foight et al. (2014)*, to a complete competitive binding model (equation 17 in *Roehrl et al. [2004]*) using a Python script.

### Library design

Position-specific scoring matrices based on SPOT array intensities ($PSSM_{SPOT}$) were described previously (*DeBartolo et al., 2012*; *Dutta et al., 2010*; *London et al., 2012*). $PSSM_{SPOT}$ scores were normalized to wild-type BIM BH3, as described by Dutta et al (*Dutta et al., 2010*). The structure-based statistical potential STATIUM was used to predict and score the effect of mutations in BH3 peptides on binding to Bfl-1, Mcl-1, Bcl-x_L (*DeBartolo et al., 2014*, *2012*). The crystal structures used to create the STATIUM models were the same as those used in previous studies (*DeBartolo et al., 2014*, *2012*): 3MQP (Bfl-1:Noxa) (*Guan et al., 2010* in press), 3PK1 (Mcl-1:BAX) (*Czabotar et al., 2011*) and 3I08 (Bcl-x_L:BIM3aF) (*Lee et al., 2009*). STATIUM z-scores were normalized using the score distribution for the human proteome, as described by *DeBartolo et al. (2014)*.

Libraries were constructed using degenerate codons chosen by a computational optimization protocol (*Chen et al., 2013*). To guide codon selection, we divided residue substitutions into three categories: *preferred*, *required* and *disruptive*. Preferred substitutions were those that scored higher than the median of all point mutants of BIM at positions 2a–4e on either $PSSM_{SPOT\_Bfl-1}$ or STATIUM_{Bfl-1}. Additionally, some substitutions that did not meet these criteria but that had large specificity scores from either $PSSM_{SPOT\_Bfl-1}$ or STATIUM_{Bfl-1} were included. Required substitutions, designated manually, were a subset of the most promising preferred residues, particularly those predicted to be highly selective for Bfl-1 or BIM/PUMA wild-type residues. Specificity for Bfl-1 over Bcl-x_L or Mcl-1 was determined by the difference of $PSSM_{SPOT}$ scores or the difference in STATIUM z-scores. Disruptive residues included mutations with $PSSM_{SPOT}$ or STATIUM scores for Bfl-1 that were more than one standard deviation worse than wild-type BIM. Degenerate codons were considered as possibilities for design if they included all the required residues at a site and none of the disruptive residues. Codons that encoded three or fewer variants were eliminated, to decrease the likelihood that a large percentage of the library would be 'poisoned' by a disruptive substitution that was not identified by our models. Combinations of degenerate codons were optimized with integer linear programming, as previously described, to maximize the number of sequences composed of preferred residues (*Chen et al., 2013*). The library was limited to at most $1 \times 10^7$ DNA sequences. The final Bfl-1 targeted library contained a large number of protein sequences ($6.84 \times 10^6$), many of which were predicted to be tight and selective Bfl-1 binders by the $PSSM_{SPOT}$ and STATIUM models. The entire design process was repeated to produce libraries selective for Bcl-x_L and Mcl-1.

### Construction of the yeast-display vector and the combinatorial library

DNA encoding PUMA-BH3 (residues 132–172 from human PUMA, UniProt # Q9BXH1-1) with a carboxy-terminal FLAG tag was subcloned into the plasmid pCTCON2 (*Chao et al., 2006*) between

Nhe1 and Xho1 restriction digest sites (5' NheI- GGTACCGGATCCGGTGGC-PUMA BH3-GGCGGCCGCGATTATAAAGATGATGATGATAAATAA-Xho1-3'). The BH3 peptide library was constructed with homologous recombination. The inserts were constructed using the PUMA-BH3 yeast display vector as a template, a reverse primer (5' CTAAAAGTACAGTGGGAACAAAGTCG 3') and forward primers with degenerate bases (PUMA Bfl-1 targeted library: 5' C GGA TCC GGT GGC CAA TGG **VHA** CGT GAA ATT **KVT** GCC **NDC** CTG CGT CGC **NBC** GCG GAT **VWK NHT** AAT GCC CAA **NYT** GAA CGT CGT CGC CAG GAG GAA C 3'; BIM Bfl-1 targeted library: 5' GGA TCC GGT GGC CGT CCG **VHA** ATT TGG ATT **KVT** CAG **NDC** CTG CGT CGT **NBC** GGC GAT **VWK NHT** AAT GCG TAT **NYT** GCG CGT CGC GTG TTT CTG AAT 3'; PUMA Bcl-x$_L$ targeted library: 5' C GGA TCC GGT GGC CAA TGG **VWS** CGT GAA **NWT** GGC GCC CAA CTG **RBA** CGC **NNC GSC** GAT GAT CTG **VHC RMA** CAA **NVC** GAA CGT CGT CGC CAG GAG GAA C 3'; BIM Bcl-x$_L$ targeted library: 5' GGA TCC GGT GGC CGT CCG **VWS** ATT TGG **NWT** GCG CAG GAA CTG **RBA** CGT **NNC GSC** GAT GAA TTT **VHC RMA** TAT **NVC** GCG CGT CGC GTG TTT CTG AAT 3'; PUMA Mcl-1 targeted library: 5' GT ACC GGA TCC GGT GGC CAA **NSG** GCG **BNC SAW RYC RBT** GCC CAA CTG **RNA** CGC ATG GCG GAT GAT **NHT VAK** GCC CAA TAT GAA CGT CGT CGC C 3'; BIM Mcl-1 targeted library: 5' TACCGGATCCGGTGGCCGT **NSG** GAA **BNC SAW RYC RBT** CAG-GAACTG**RNA** CGTATTGGCGATGAA **NHT VAK** GCGTATTATGCGCGTCGCGT 3'). To complete insert construction, the 5' ends of these PCR products were further extended until there was at least 40 bp of homology to the acceptor vector on both ends of the library inserts. The acceptor vector was prepared by cleaving the yeast display vector with the endonucleases Xho1 and Nhe1 (NEB, Ipswich, MA) and purifying the cleavage product with a gel extraction kit (Qiagen, Hilden, Germany). The library inserts and acceptor vector were mixed and transformed into yeast following the procedure of *Gietz and Woods (2002)*. Twenty electroporations produced >10-fold more transformants than the theoretical size of each library with vector background estimated at <0.01%. DNA from transformed cells was PCR amplified to check for randomization.

## Flow cytometric analysis and sorting

The yeast-displayed Bfl-1 library was grown and sorted using fluorescence-activated cell sorting (FACS) according to a protocol adapted from *Reich et al., 2016*). The libraries were grown from glycerol stocks that were inoculated to a final OD$_{600}$ of 0.05 in a volume sufficient to oversample the estimated library diversity by at least 10-fold in selective media containing glucose (SD+CAA: 5 g/L casamino acids, 1.7 g/L yeast nitrogen base, 5.3 g/L ammonium sulfate, 10.2 g/L Na$_2$HPO$_4$-7H$_2$O and 8.6 g/L NaH$_2$PO$_4$-H$_2$O, 2% glucose). Cultures were grown for 12 hr at 30°C and then cells were diluted to OD$_{600}$ of 0.005–0.01 in SD+CAA and grown to OD$_{600}$ of 0.1–0.6 (~12 hr) at 30°C. To induce expression, cultured were diluted (40 mL inoculate/L) into selective media containing galactose (SG+CAA: 5 g/L casamino acids, 1.7 g/L yeast nitrogen base, 5.3 g/L ammonium sulfate, 10.2 g/L Na$_2$HPO$_4$-7H$_2$O and 8.6 g/L NaH$_2$PO$_4$-H$_2$O, 2% glucose) and grown to OD$_{600}$ of 0.2–0.5 (16–24 hr) at 30°C. Induced yeast cells were filtered with 0.45 μm filter plates or bottle-top filters and washed twice with BSS (50 mM Tris, 100 mM NaCl, pH 8, 1 mg/ml BSA). Sufficient cells to oversample the library diversity at least 10-fold were resuspended in BSS with at least 10-fold molar excess target protein and incubated for 2 hr at room temperature with gentle shaking. Cells were filtered, washed twice in chilled BSS and incubated with a mixture of primary antibodies (anti-HA mouse, Roche, Indianapolis, IN, RRID:AB_514505, and anti-c-myc rabbit, Sigma, St. Louis, MO, RRID:AB_439680) at 1:100 dilution in a volume of 20 μl per 10$^6$ cells for 15 min at 4°C in BSS. Cells were filtered, washed twice in chilled BSS and incubated in a mixture of secondary antibodies (1:40 APC rat anti-mouse, BD, San Jose, CA, RRID:AB_398465, and 1:100 PE goat anti-rabbit, Sigma, RRID:AB_261257) in BSS at 4°C in the dark for 15 min. The filtering and washing steps were repeated and the labeled cells were resuspended in BSS and analyzed on a BD FACSCanto flow cytometer or sorted on a BD FACSAria using FACSDiva software. The sorted cells were collected in selective media containing glucose (SD+CAA) and grown to an OD$_{600}$ of 6–10 for ~48 hr in the presence of streptomycin/penicillin to prevent bacterial growth, then pelleted, washed and stored as glycerol stocks in SD+CAA + 20% glycerol. A series of positive, negative and competition sorts were used to enrich Bfl-1 selective binders. The detailed sorting scheme is given in *Figure 2—figure supplement 1*.

## Illumina sequencing and data processing

Glycerol stocks from each pool isolated during sorting were grown overnight in SD+CAA, using sufficient stock to oversample the estimated library diversity by at least 10-fold. $1 \times 10^8$ cells from each pool were pelleted in a microcentrifuge tube at $300 \times$ g for 1 min and washed twice with PBS. The plasmid DNA from yeast was extracted using the Zymoprep Yeast Plasmid Miniprep II (Zymo Research, Irvine, CA) reagents and Qiagen miniprep columns. The DNA was eluted in water. The BH3 library was amplified with PCR using primers that encoded an MmeI restriction enzyme binding site at 5′ end and a universal Illumina sequencing region on the 3′ end. After purification with the Qiagen PCR purification kit, the PCR products were digested with MmeI (3.45 pmol DNA:2 μL MmeI, NEB) for 1 hr at 37°C before being heat inactivated at 80°C for 20 min. Each digestion product was then ligated by treatment with T4 DNA ligase (NEB) for 30 min at 20°C to double-stranded DNA fragments containing Illumina adapters, with the adapter containing a unique barcode, and heat inactivated for 10 min at 65°C. Barcodes were varied by at least two bases and were used to assign Illumina reads to the appropriate pool. A final PCR amplified the ligation product and extended the 5′ and 3′ regions to include adaptor sequences for Illumina sequencing. Samples were then multiplexed and run in one lane on an Illumina Nextseq with paired-end reads of 75 bp using the universal Illumina forward sequencing primer and a PUMA construct-specific Illumina read primer reverse (5′ CGCCTTGTTCCTCCTGGCGACGACGTTCATATTGGGC 3′). Illumina deep sequencing data was processed in python. The data were filtered for sequences with barcodes that had high Phred Scores (>20). Sequences were reconstructed by aligning pair end reads. Sequences observed fewer than 20 times were removed from the data set.

## Crystallography

Crystals of Bfl-1 in complex with PUMA, FS2 or FS2_1fX peptides were grown in hanging drops over a reservoir containing 1.8 M ammonium sulfate, 0.1 M MES pH 7.0 at room temperature. Crystals were seeded with drops containing parent crystals grown in higher ammonium sulfate (2.2–2.4 M) using a cat whisker. The protein was mixed with peptide at a 1:1 molar ratio and concentrated to 4 mg/mL in 20 mM Tris, 150 mM NaCl, 1% glycerol, 1 mM DTT, pH 8.0. The hanging drops contained 1.5 μL of complex mixed with 1.5 μL of reservoir solution. Crystals were cryo-protected by transferring into 2.0 M lithium sulfate with 10% glycerol prior to flash freezing. Diffraction data were collected at the Advanced Photon Source at the Argonne National Laboratory, NE-CAT beamline 24-ID-C. The Bfl-1:FS2 data were integrated and scaled to 1.2 Å using HKL2000 and phased using PHENIX ridged body refinement of chain A of structure 4ZEQ using PHENIX (*Otwinowski and Minor, 1997*; *Adams et al., 2010*). The peptide was built into the difference density from the rigid body refinement and the structure was refined with iterative rounds of refinement and model building using PHENIX and COOT (*Adams et al., 2010*; *Emsley et al., 2010*). The PUMA and FS2_1fX complex data sets extended to 1.33 Å and 1.73 Å, respectively, and were phased with the Bfl-1 chain of the FS2 complex model (*McCoy et al., 2007*).

Crystals of the Mcl-1/FS2 peptide complex were grown at room temperature in hanging drops over a reservoir containing 0.2 M zinc sulfate, 0.1 M imidazole (pH 6.5), and 3% 6-aminohexanoic acid. The protein was mixed with peptide at a 1:1 molar ratio and diluted to 2 mg/ml in 20 mM Tris, 150 mM NaCl, 1% glycerol, 1 mM DTT, pH 8.0. The hanging drops contained 1.5 μL of complex mixed with 1.5 μL of reservoir solution. Crystals were cryoprotected by transferring into 15% glycerol, 0.2 M zinc sulfate, 0.1 M imidazole (pH 6.5) and 3% 6-aminohexanoic acid prior to flash freezing. Diffraction data were collected at the Advanced Photon Source at the Argonne National Laboratory, NE-CAT beamline 24-ID-E. The data were processed to 2.35 Å and phased using molecular replacement with chain A of structure 3PK1 (*Czabotar et al., 2011*) using PHASER and refined using PHENIX and COOT (*Otwinowski and Minor, 1997*; *Adams et al., 2010*; *Emsley et al., 2010*; *McCoy et al., 2007*).

## Gel shift assays

Myc-tagged Bfl-1 (5 μM) was incubated with BH3 peptide (25 μM) in 200 μL of FP in 200 μL FP buffer (see above). 20 μL subsamples were taken at 0, 1, 2, 4, 8, 16, 32, 64 and 128 min and quenched in 7 μL loading buffer. Samples were immediately flash frozen in liquid nitrogen and stored at −80°C. Samples were run on a 14% acrylamide SDS-PAGE gel and visualized with Coomassie Brilliant Blue.

Bands were quantified with ImageJ. Data were fit in python to the equation $y=C*(1-e^{-kt})$, where y is the fraction of cross-linked Bfl-1, C is the upper limit, t is time and k is the decay constant.

## Cell lines

The creation and characterization of the BCR-ABL-expressing B-lineage acute lymphoblastic leukemia suspension cell lines with engineered dependencies on human versions of anti-apoptotic genes is detailed in Koss et al. (2016)). Cells were grown in RPMI (Life Technologies, Carlsbad, CA) with 10% fetal bovine serum, 2 mM L-glutamine, 10 mL/L 100x Pen/Strep (Life Technologies # 15140122), 25 mM HEPES and 10 mL/L 100x NEAA (Life Technologies, 11140050). The adherent cell lines PC-3 (RRID: CVCL_0035) and SF295 (RRID: CVCL_1690) are from the NCI60 panel (Lorenzi et al., 2009) and were grown in RPMI (Life Technologies) with 10% fetal bovine serum, 2 mM L-glutamine and 10 mL/L 100x Pen/Strep (Life Technologies # 15140122 ). Cell line identities were confirmed by STR profiling. The Lookout Mycoplasma PCR detection kit (Sigma) was used to detect mycoplasma infection. Mycoplasma was only detected in the PC-3 cell line, and internal controls were used to account for this phenotype.

## BH3 profiling assays

Peptides were titrated by serial dilution in MEB buffer (150 mM Mannitol, 10 mM HEPES-KOH pH 7.5, 50 mM KCl, 0.02 mM EGTA, 0.02 mM EDTA, 0.1% BSA and 5 mM Succinate) containing 20 μg/mL oligomycin, 50 μg/mL digitonin, 2 μM JC-1 and 10 mM 2-mercaptoethanol in 384-well plates. Controls for no depolarization (1% DMSO) and complete depolarization with the mitochondrial oxidative phosphorylation uncoupler FCCP (20 μM) were included for data normalization. Cells were suspended at $1.67 \times 10^6$ cells/mL in MEB. 15 μL of cell suspension was added to each well containing 15 μL of treatment solution. Fluorescence emission was measured every 5 min for 3 hr at 590 nM with 525 nM excitation on a Tecan Safire2. To produce percent depolarization, the area under the resultant curve was calculated and normalized to the assay controls. Peptide titration curves were fit to sigmoidal dose-response curves using Graphpad PRISM seven to obtain $EC_{50}$ values.

## iBH3 assays

Cells were suspended in MEB buffer (150 mM mannitol, 50 mM KCl, 10 mM HEPES, 5 mM succinic acid, 20 μM EGTA, 20 μM EDTA, 0.1% BSA, final pH 7.4) at $0.5*10^6$ cells/mL (adherent lines) or $2*10^6$ cells/mL (suspension lines). Cell suspension was added to a 384 non-binding well plate (10 μL/well) containing peptides at 2X final concentration in MEB with 20 μg/mL digitonin. Plates were incubated at 25°C for 1 hr. To terminate exposure, 10 μL of 4% formaldehyde in PBS was added to each well, plates were incubated for 10 min before addition of 10 μL N2 buffer (1.7M Tris, 1.25M glycine, pH 9.1) for 5 min. 10 μL of staining buffer (2% Tween20, 10% BSA, PBS) containing 10 μg/mL Hoechst 33342 and 1.25 μg/mL anti-cytochrome c Alexa647 conjugate (BioLegend clone 6H2.B4) was added to each well before sealing the plate and shaking overnight. The median fluorescence of the cytochrome c channel of Hoechst positive singlets was recorded by an IntelliCyt iQue Screener Plus. Cytochrome c release was determined by normalizing the median fluorescence intensity (MFI) data to positive control wells (Alamethicin) and negative control wells (DMSO) as follows:

$$\mathrm{Cytochrome\,c\,release} = 1 - (MFI_{sample} - MFI_{Alamethicin})/(MFI_{DMSO} - MFI_{Alamethicin})$$

## Acknowledgements

We thank the Koch Institute Bioploymers and Proteomics Facility for peptide synthesis, the Koch Institute Flow Cytometry Core Facility for assistance with FACS sorting, and the MIT Structural Biology Core Facility for assistance with X-ray crystallography. We thank LRF Backman for help with x-ray data collection. We thank members of the Keating lab, especially T Hwang for performing gel shift assays, R Rezaei Araghi for performing mass spec, V Xue for help processing Illumina sequencing reads, V Frappier for help with structural analysis, and G Foight for help with library design. We thank the laboratory of J Opferman for providing the cell lines of defined anti-apoptotic dependence used in the BH3 profiling assays. Part of this work was conducted at the Northeastern Collaborative Access Team beamlines, which are funded by the National Institute of General Medical Sciences from the National Institutes of Health (P41 GM103403). The Pilatus 6M detector on 24-ID-C beam line is funded by a NIH-ORIP HEI grant (S10 RR029205). This research used resources of the

Advanced Photon Source, a U.S. Department of Energy (DOE) Office of Science User Facility operated for the DOE Office of Science by Argonne National Laboratory under Contract No. DE-AC02-06CH11357. The content is solely the responsibility of the authors and does not necessarily represent the official views of the National Institutes of Health or the U.S. Department of Energy.

## Additional information

### Funding

| Funder | Grant reference number | Author |
|--------|------------------------|--------|
| National Institute of General Medical Sciences | R01-GM110048 | Justin M Jenson<br>Amy E Keating |
| National Science Foundation | Graduate Student Fellowship | Justin M Jenson |
| MIT School of Science Ludwig Fund for Cancer Research | Graduate Student Fellowship | Justin M Jenson |
| Koch Institute/MIT - Dana-Farber/Harvard Cancer Center Bridge Project | | Jeremy A Ryan<br>Anthony Letai |

The funders had no role in study design, data collection and interpretation, or the decision to submit the work for publication.

### Author contributions

JMJ, Conceptualization, Data curation, Formal analysis, Investigation, Visualization, Writing—original draft; JAR, Data curation, Formal analysis, Investigation, Writing—review and editing; RAG, Resources, Formal analysis, Supervision, Investigation, Writing—review and editing; AL, Resources, Supervision, Funding acquisition, Writing—review and editing; AEK, Conceptualization, Supervision, Funding acquisition, Writing—review and editing

### Author ORCIDs

Justin M Jenson, http://orcid.org/0000-0003-1960-0447
Jeremy A Ryan, http://orcid.org/0000-0002-3327-1283
Amy E Keating, http://orcid.org/0000-0003-4074-8980

## Additional files

### Supplementary files

• Supplementary file 1. Summary of X-ray data collection and refinement statistics.

### Major datasets

The following datasets were generated:

| Author(s) | Year | Dataset title | Dataset URL | Database, license, and accessibility information |
|-----------|------|---------------|-------------|--------------------------------------------------|
| Jenson JM, Keating AE | 2017 | Epistatic mutations in PUMA BH3 drive an alternate binding mode to potently and selectively inhibit anti-apoptotic Bfl-1 | https://www.ncbi.nlm.nih.gov/geo/query/acc.cgi?acc=GSE94864 | Publicly available at the NCBIGene Expression Omnibus (accession no: GSE94864) |
| Jenson JM, Grant RA, Keating AE | 2017 | Human Bfl-1 in complex with a Bfl-1-specific selected peptide | http://www.rcsb.org/pdb/explore/explore.do?structureId=5UUK | Publicly available at the RCSBProtein Data Bank (accession no: 5UUK) |
| Jenson JM, Grant RA, Keating AE | 2017 | Human Bfl-1 in complex with Puma BH3 | http://www.rcsb.org/pdb/explore/explore.do?structureId=5UUL | Publicly available at the RCSBProtein Data Bank (accession no: 5UUL) |

| Jenson JM, Grant RA, Keating AE | 2017 | Human Mcl-1 in complex with a Bfl-1-specific selected peptide | http://www.rcsb.org/pdb/explore/explore.do?structureId=5UUM | Publicly available at the RCSBProtein Data Bank (accession no: 5UUM) |
|---|---|---|---|---|
| Jenson JM, Grant RA, Keating AE | 2017 | Human Bfl-1 covalently cross-linked to an electrophilic variant of a Bfl-1-specific selected peptide | http://www.rcsb.org/pdb/explore/explore.do?structureId=5UUP | Publicly available at the RCSBProtein Data Bank (accession no: 5UUP) |

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
