## [Decision Letter]

Thank you for submitting your article "Epistatic mutations in PUMA BH3 drive an alternate binding mode to potently and selectively inhibit anti-apoptotic Bfl-1" for consideration by *eLife*. Your article has been favorably evaluated by John Kuriyan (Senior Editor) and three reviewers, one of whom, Mingjie Zhang (Reviewer #1), is a member of our Board of Reviewing Editors. The following individual involved in review of your submission has agreed to reveal his identity: Gaetano T Montelione (Reviewer #3).

The reviewers have discussed the reviews with one another and the Reviewing Editor has drafted this decision to help you prepare a revised submission. You will see that the reviewers have raised some issues that will require additional experiments.

Summary:

Jenson et al. have designed specific BH3-like peptide inhibitors of anti-apoptotic Bf1-1, an important BCl^-^2 protein that regulates apoptosis. Bfl-1 is also an important target for development of novel cancer therapeutics. The peptide design process includes both sequence-based and structure-based virtual screens, followed by iterative yeast peptide-display screening, synthesis of specific target peptides, and extensive characterization of specificity and affinity, using both permeablized cell-based mitochondrial membrane disruption assays and biophysical measurements of peptide – BCl^-^2 protein binding affinities. Following on studies reported by other groups, PUMA-like peptides which can form covalent bonds with residue Cys-55 in the Bfl-1 BH3-peptide binding site, were also designed. The structure of one of these covalent peptide – protein complexes was also solved by X-ray crystallography.

A key strength of this work is the demonstration of an effective and powerful combination of large scale computational modeling (an approach that can deal with sequence spaces that are beyond the limits of bench experiments) and experimental validation and optimization (an approach that is critical to achieve necessary accuracy, currently not easily achievable by pure computational methods) in discovering highly specific and potent peptide binders. Using the Bfl-1/PUMA BH3 complex structure as the input model, the authors were able to reduce the sequence space of the BH3-derivitive peptide library to an experimentally accessible number. They further used 3D structures of various BH3 peptide – BCl^-^2 complexes for both positive and negative designs, in order to create high specificity and to minimize off-target binding without significantly destabilizing Bfl-1 binding. X-ray crystallography reveals that the designed peptide inhibitors use a binding mode that is distinct from that observed in other BH3 peptide – Bfl-1 complexes, providing insight into the structural basis of the observed specificity. This novel binding mode structure was then used to more accurately score candidate inhibitor peptides using structure-based virtual screening.

In addition to developing highly specific Bfl-1 binding peptides, this study also presents significant methodological advances for designing selective peptide inhibitors, which will be broadly applicable to other systems. As such, the work is a strong candidate for publication in *eLife*.

Important points to address:

1) Regarding the validation of specificity of the peptides, the authors should evaluate the FS peptides for targeting Bfl-1 protein and confirm they do not target other BCl^-^2 proteins in cell extracts using a direct measurement, such as a pull-down experiment, as the depolarization assay is an indirect readout. They should also evaluate binding with cell extracts and fluorescence polarization to pro-apoptotic BCl^-^2 proteins (Bax and Bak) using c-terminal helix truncated constructs since PUMA BH3 binds and activates pro-apoptotic Bax and Bak.

2) Figure 5. A key difference appears to be in the Asp at the position 3f. The structure has provided a compelling explanation for the binding difference between the FS2 peptide and the PUMA peptide in binding to Bfl-1. In the FS2 peptide, D3f does not form the critical D3f:R88:D81 network. However, it is rather puzzling why D3f is recovered at a 100% rate in the final library as shown in Figure 4. Based on the structure shown in Figure 5, Asp at the 3f position of the FS2 peptide may be tolerated by substitutions of other residues. The authors need to test this possibility by a mutational approach.

3) The combination of computational design and yeast-display library screening is a strength of this approach. The reviewers wonder to what extent accidental mutations played a role in optimizing the PUMA peptide and not the BIM peptide, as alluded to in the text. The authors should comment on the extent to which adventitious mutations may be an important component of the optimization strategy going forward.

---

## [Author Response]

*Important points to address:*

*1) Regarding the validation of specificity of the peptides, the authors should evaluate the FS peptides for targeting Bfl-1 protein and confirm they do not target other BCl^-^2 proteins in cell extracts using a direct measurement, such as a pull-down experiment, as the depolarization assay is an indirect readout.*

To demonstrate specificity, we used two complementary methods. The first was a direct, quantitative fluorescence polarization (FP) binding assay that we performed in vitro using purified proteins (Figure 2). Biophysical assays like this (using FP or surface plasmon resonance) are the gold standard in the field for demonstrating binding specificity (e.g. see Opferman, 2015; Souers et al., 2013; Roberts et al., 2016; Rudin et al., 2012; Roberts et al., 2012). Our second assay was a functional assay (BH3 profiling), which used permeabilized cells and measured the key end point of the BCl^-^2 mediated process we are studying: mitochondrial depolarization caused by mitochondrial outer membrane permeabilization (MOMP).

Our quantitative FP assay provides a more direct assessment of binding specificity than would a pull-down from lysates, and the BH3 profiling assay is a superb probe of function in a cellular context (see e.g. Opferman, 2015; Roberts et al., 2012; Schoenwaelder et al., 2011). It is not clear what the suggested pull-down experiments would add, on top of the existing characterization, that would be necessary to support our key claims of specificity of *binding* and of *function*. An important challenge of interpreting pull-down experiments is that treatment of lysates with a highly specific Bfl-1 directed peptide could still cause displacement of pro-apoptotic proteins from other anti-apoptotics indirectly, as the pro-apoptotics displaced by the peptide would be free to compete for binding at the other anti-apoptotics. Furthermore, the proposed pull-down experiments would take at least 3-4 months to develop, optimize and perform to publication standards, given that we would need to identify and obtain suitable panels of cells that express all BCl^-^2 paralogs at appropriate levels, and identify sufficiently clean antibodies for the required IPs and blotting. These results would not provide any critical missing data to support our claims.

Summary: the reviewers indicated that a more direct readout of specificity than the BH3 profiling assay is required. We have provided this in the form of quantitative K_i_ values for all peptides binding to all anti-apoptotic proteins (Figure 2—figure supplement 4).

*They should also evaluate binding with cell extracts and fluorescence polarization to pro-apoptotic BCl^-^2 proteins (Bax and Bak) using c-terminal helix truncated constructs since PUMA BH3 binds and activates pro-apoptotic Bax and Bak.*

As the reviewer points out, PUMA is reported to function as an activator of BAK/BAX. It has also been observed that, under certain conditions (absence of lipid/detergent), PUMA can form a (relatively weak) complex with BAX and BAK (see Opferman, 2015; Ryan, Brunelle and Letai, 2010; Deng et al., 2007). To our knowledge, no one has shown or suggested that PUMA stably binds to native BAK or BAX in cells. Thus, the key question is whether our peptides, which are based on PUMA, retain the BAK/BAX activating function of PUMA. We agree that this is important to address.

We previously included data for several cell lines in which BAK and/or BAX were present yet our FS1-3 peptides did not induce MOMP. This contrasts with the behavior of the activator peptide BIM BH3 in these cells, arguing against a role for FS1-3 as direct activators (Figure 7).

To further test for activator function of FS1, FS2 and FS3, we performed experiments to measure activation in “unprimed” cells. Such cells require an activator to induce MOMP. We measured cytochrome c release in response to treatment of permeabilized cells with peptides. The new data, shown in Figure 7—figure supplement 2, are clear. Whereas both BIM BH3 (a known activator) and the PUMA BH3 peptide on which our designs are based led to concentration-dependent release of cytochrome c in unprimed cells, designed peptides FS1 and FS2 showed no activity up to a concentration of 100 μM (highest concentration tested).

While running these experiments, we also tested for cytochrome c release in primed cells with different BCl^-^2 dependencies, as an additional measure of functional specificity. Consistent with our previous data, FS1, FS2 and FS3 released cytochrome c from mitochondria in permeabilized Bfl-1 dependent cells at concentrations < 10^-8^ M. In contrast, the concentrations required for release in cell lines dependent on BCl^-^x_L_, BCl^-^2 or MCl^-^1 were at least 3 orders of magnitude greater (Figure 7—figure supplement 1).

We did learn from these experiments that FS3 leads to cytochrome c release at high concentrations in unprimed cells, consistent with it having some weak activator activity (Figure 7—figure supplement 2). But the concentration of FS3 at which cytochrome c is released in unprimed cells is 3-4 orders of magnitude higher than the concentration required to induce MOMP in Bfl-1 dependent primed cells (Figure 7—figure supplement 1).

We are excited about the data from these new experiments, which we performed in response to the reviewers’ comments. The results support and strengthen our conclusions.

Summary: It is important to establish whether our designed peptides function as activators of BAK or BAX, given that PUMA (on which our peptides are based) can have this function. Because the relationship between binding and activation is not established, activation is best tested using a functional assay. Our several tests confirm that FS1 and FS2 do not have activator activity, at least up to concentrations two orders of magnitude higher than the concentrations at which PUMA acts as an activator, and at least 4 orders of magnitude above the concentration at which we see selective MOMP induced by our peptides in Bfl-1 dependent cell lines.

The following figures have been added or modified:

Figure 7 – bar plot of cytochrome c release data showing specificity;

Figure 7—figure supplement 1, cytochrome c release titrations performed in our panel of BCl^-^2 over-expressing cell lines;

Figure 7—figure supplement 2. Peptide function in unprimed cells PC-3 and SF295, showing FS1 and FS2 are not activators like BIM and PUMA.

Modified text to describe the additions is included in the subsection “Biological Activity of Designed Bfl-1 inhibitors”.

*2) Figure 5. A key difference appears to be in the Asp at the position 3f. The structure has provided a compelling explanation for the binding difference between the FS2 peptide and the PUMA peptide in binding to Bfl-1. In the FS2 peptide, D3f does not form the critical D3f:R88:D81 network. However, it is rather puzzling why D3f is recovered at a 100% rate in the final library as shown in Figure 4. Based on the structure shown in Figure 5, Asp at the 3f position of the FS2 peptide may be tolerated by substitutions of other residues. The authors need to test this possibility by a mutational approach.*

All of our peptides have aspartate at position 3f because this site was not varied in our libraries (Figure 1—figure supplement 2). Thus, there is no mystery about why this residue was recovered. Our structure does suggest that substitution of this highly conserved residue should be possible at this site in the altered binding geometry, and we have now tested this hypothesis. Figure 5—figure supplement 2 shows that substitution of aspartate at 3f with any of alanine, serine, asparagine, glutamate, histidine or tyrosine is well tolerated, as anticipated.

Summary: It is not puzzling why aspartate at 3f was recovered at 100%, because it wasn’t varied. But substitution of diverse residues at this position is indeed tolerated, as demonstrated by our new data.

The following figure was added:

Figure 5—figure supplement 2. Yeast displayed FS2 point mutants.

These data are referenced in the fourth paragraph of the subsection “The binding mode of Bfl-1-selective peptides”.

*3) The combination of computational design and yeast-display library screening is a strength of this approach. The reviewers wonder to what extent accidental mutations played a role in optimizing the PUMA peptide and not the BIM peptide, as alluded to in the text. The authors should comment on the extent to which adventitious mutations may be an important component of the optimization strategy going forward.*

We think it unlikely that accidental mutations differentially affected BIM vs. PUMA peptides, because a strong preference for PUMA variants over BIM variants was evident early in screening (in pool FL2, see Figure 2—figure supplement 1). Deep sequencing of clones from the unsorted library, from both the BIM and PUMA libraries, showed that both naive libraries contained low, and very similar, frequencies of non-designed mutations. Furthermore, this increased very little for the PUMA library between the naïve library and the FL3 pool (from 4.4% to 5.9%). Thus, the BIM library defect (which was dramatic, see Figure 2—figure supplement 2) appeared at a point in the experiment where the libraries were diverse and overwhelmingly contained designed sequences.

We don't believe that adventitious mutations were critical for the success of the PUMA library screen. Among our three “winner” peptides, a single un-designed mutation was present in peptide FS1; FS2 and FS3 were encoded in the designed library. The FS1 mutation is at a solvent exposed position (R3cL). Based on the PUMA and FS2 structures in Figure 5, it seems unlikely that a leucine at this position would have much effect on Bfl-1 binding. Further, this position is one of only 2 sequence differences between FS1 and FS2, the other being a fairly conservative mutation (L to V mutation at position 4e), and yet FS1 and FS2 have similar binding affinities.

Added: Discussion, third, fourth and eighth paragraphs.